# Improving Operational and Sustainability Performance in a Retail Fresh Food Market Using Lean: A Portuguese Case Study

Pedro Alexandre Marques [1,*], André M. Carvalho [2] and José Oliveira Santos [1]

1. EIGeS—Research Centre in Industrial Engineering, Management and Sustainability, Lusófona University, Campo Grande, 376, 1749-024 Lisboa, Portugal; jose.oliveira.santos@ulusofona.pt
2. Polytechnic Institute of Cávado and Ave, 4750-810 Funchal, Portugal; andrecarvalho@ipca.pt
* Correspondence: pedro.alexandre.marques@ulusofona.pt; Tel.: +351-21-751-5500

**Abstract:** This paper presents a real application of a lean–green improvement initiative conducted at a large Portuguese hypermarket store. It explores how lean tools and techniques may be used to not only improve the operational performance, but also sustainability. A case study was carried out in one store of a multinational retail enterprise, with the aim of enhancing both the operational and sustainability performance in the cold meat section, one of the most relevant areas of the fresh food markets. The Gemba Kaizen event approach, which comprises three main stages, was adopted. During the workshop stage, the structured problem-solving methodology was followed, and was recorded in an A3 format. As a consequence of this project, food waste in the cold meat market was reduced by half, whereas the out-of-stock index decreased by a third. In addition, the pilot store hit top performance within all stores of the company in Portugal, ranking first in all key indicators for the cold meat market. The lean–green scope and performance improvement procedures developed and implemented in the pilot store were later deployed to other stores of the company. This is one of the first publications regarding the application of lean management in the food retail sector for improving both the operational and sustainability performance.

**Keywords:** food waste; fresh products markets; lean; out-of-stocks; shrinkage; sustainability

## 1. Introduction

Food retailers offer a wide range of fresh products, including fruits and vegetables, meat, fish, baked goods, or cold meats and cheese, among others. Fresh categories typically account for up to 40 percent of grocery chains' revenues and, in addition to this, they are also strong drivers of store traffic and customer loyalty [1]. However, they also represent a major risk, derived from lost sales, food waste, and the resulting perception of unsustainable operations.

Retail companies continuously face stiff competition, a reduction in sales margins, and increasingly demanding customers [2]. At the same time, regarding sustainability, the impact of fresh food waste is immense. Retailers, governments, and associations worldwide have taken different approaches to curb the problem. Price reductions or offerings of fresh produce, legislation preventing the dumping of food perishables, and the development of new business models are some the solutions that have risen in the face of this problem [3–5]. Several examples can be provided. Food producers and retailers have redesigned labels and packaging in an effort to prevent loss and to avoid food waste [6,7]. Countries forbade food retailers from dumping fresh food products [8]. Associations, communities, and start-ups around the world are coming up with new solutions to reduce food waste [3,9].

Accordingly, the performance of fresh food departments and the reduction in waste are two major objectives of food retailers and markets in the present day [10]. It has been argued that the performance of these markets can be improved by improving freshness, increasing sales, or reducing "shrinkage" [11]. In a fresh food market, the shrinkage rate

is able to measures the proportion of food wasted due to spoilage, mishandling, a loss of quality, and the expiration date, thus also leading to the loss of potential sales of a store. From a sustainability point of view, the term "shrinkage" is key, as it does not demand further incentives to consumption or require faster transportation and/or further energy usage. Furthermore, from an operational point of view, the focus on shrinkage has a benefit. Being seen as "waste", its reduction is easily framed within traditional operational improvement initiatives already used in the industry, such as lean operations. In fact, shrinkage has been loosely applied to encapsulate some of the areas that generate loss [12].

The benefits of matching lean and sustainability—or "green"—improvement initiatives may be explored from multiple perspectives. There have been some approaches joining the operational and sustainability perspectives in the retail sector, but the research trend is still emerging [13]. Examples include the improved use of technology [14], the inclusion of shrinkage as a variable for optimization [15], or the promotion of sustainability-oriented management [16,17]. However, clear descriptions of the methods used to join the sustainability and operational performance improvement in the retail sector—especially those based on real cases—are still scarce.

In this work, we focus on "shrinkage" as an indicator of sustainability, once it is applied in a fresh food market with the intention to measure and reduce food waste. By being traditionally recognized within the lean practices of the industry (usually to gauge lost sales), we expect that it will more easily be accepted by both leaders and the workforce. By using a familiar lean indicator to tackle sustainability issues, we hope to align the pursuit of the "green" perspective with that of the operational performance, reducing misconceptions and resistance to sustainability initiatives.

In light of this reality, this paper aims to understand if and how lean tools and methods may be used to efficiently tackle sustainability issues related to food waste, in the same way that they provided an effective approach to tackle operational and process waste in the retail sector. It presents a real lean–green project conducted at a Portuguese food retail store that aimed to reduce food shrinkage and the resulting waste in one of the most important fresh products markets: the cold meat section. In this paper, we focus on shrinkage events that can lead to food waste, thus not including events of theft, nor accounting errors. As a result, it was observed that these methods were tackling shrinkage in order to improve the sustainability and operational performance indicators in the fresh food market. While it aimed primarily to reduce food waste, it also intended to decrease the number of out-of-stocks (OOS) in the same market—an objective that was initially perceived as a possible trade-off, but was later seamlessly integrated within the first one.

The methodology used in this work was based on the use of traditional lean tools and methods. Three stages comprising the Gemba Kaizen event approach were adopted by the team to perform the improvement initiative. An A3 problem-solving report provided the structured roadmap to record and display all steps of the improvement cycle. As a result, shrinkage was reduced, and although some variability has been observed, the weekly average has been cut to almost half (from 4.4% to 2.4%). At the same time, OOS events were also consistently reduced.

This work provides important insights for retail managers in pursuit of a balance between sustainability and operational strategies. It shows how the use of process and operations improvement methods, especially in the scope of a lean management program, can deliver effective solutions to reduce food waste while still providing operational improvements. The article is organized around five sections. After a careful and extensive literature review about the retail industry, related sustainability issues, and the adoption of lean practices in the sector, the case is presented and described, and is followed by the discussion of the results and a summary of the conclusions. Finally, the limitations of the research are identified and suggestions for future research are proposed.

## 2. Literature Review

### 2.1. Retail Store Operations: Research Summary and Opportunities

The retail sector comprises an important portion of the economy [18]. A report issued by the European Physical Society [19] concludes that retail made a net contribution of around 4.5% to the European Union GDP, while the Price Waterhouse Coopers [20] reports that, in the United States, the direct impact of retail in the national total GDP is 7.7%. It is thus natural that the interest on the part of the academic community in this sector has increased over the years [21], as evidenced by the substantial increment of the number of papers published [22], especially from 2015 [23].

Research on retail management encompasses many topics [24], including, among others, shelve replenishment, inventory management, food waste management, product promotions, product positioning, pricing, or online commerce. Given the wide variety of themes, Caro et al. [25] propose a classification of the topics representing the central challenges in retail management, considering eight categories: (1) Inventory, (2) Pricing, (3) Assortment, (4) Incentives, (5) Online retail, (6) Industry Studies, (7) Returns, and (8) Other topics.

Similarly, Mou et al. [18] provide a complete review of the current state of research on retail store operations using a classification of six research items/themes and seven operational decisions. In their literature review, the authors classified a total of 255 journal articles, published from 2008 to 2016. The authors concluded that more than two thirds of all papers fall in the "Inventory Management Decision" category, and, within this category, that more than 95% focus on the themes of "Uncertainty", "Perishability", and "Availability".

A review of these themes quickly and clearly establishes their importance for retail operations management. The theme of "Uncertainty" comprises three perspectives: purchase quantity, purchase timing, and purchase preferences of customers. According to Bouza-abia et al. [26], the in-store operations performance significantly contributes to reducing the impact of uncertainty in customer-perceived indicators. Reducing uncertainty thus positively impacts both perishability and availability. Several studies have been made on how to manage uncertainty. Among other relevant findings, it is curious to note that high inventory levels make in-store logistics more prone to execution errors, causing a higher risk for product shrinkage [27] and shelf stockouts [28].

In fresh products categories, "Perishability" is the biggest concern of food retailers, not only because it negatively affects the customer perception of quality, but also because of the monetary loss it creates [15]. Perishable products account for over 40% of sales in grocery chains [1], and, with factors such as a limited lifetime, high safety and quality requirements, and short lead time, they are requirements that are highly complex to manage [29].

Finally, "Availability" is key among all customer-perceived indicators of customer service in retailing, including in the online channels [30]. Product availability, or on-shelf availability (OSA), is defined as the probability of having a product in stock when a customer order arrives [31]. The higher the stock level on the shelves, the lower the likelihood of an out-of-stock (OOS) situation; nevertheless, this would lead to an increase in the capital holding costs, as well as the risk for product shrinkage [27]. According to Aastrup and Kotzab [32], both retailers and producers suffer significant losses due to poor OSA. They identify two main research streams in the area of OOS: demands issues and supply issues. Both are described below:

- Demand side issues, including the study of consumer responses to stockouts [33–38]. These issues have been proven to affect store the image and brand loyalty [34]; and it has been concluded that more than 15% of customers usually decide to quit their purchase and go elsewhere to purchase a stockout product [39];
- Supply side issues, including the analyses of root-causes that explain the occurrence of OOS situations, as well as of countermeasures to improve the performance [35,40–45]. Reasons that contribute to OOS events include situations of a poor in-store operations performance [35], poor backroom inventory handling procedures [42], misplaced SKUs [43], late delivery by a supplier [44], large product variety [28], and promotional events

that lead to more uncertainty on demand [45]; some authors also claim that higher levels of the inventory make in-store logistics more prone to execution errors [28,39].

Given the current business environment, it is no surprise that "Uncertainty", "Perishability", and "Availability" rank as the top themes in the literature on retail operations management. However, and looking at the research published until 2016, the low frequency of the topic of "Sustainability" stands out. It is true that the topic has grown considerably in the past few years, with themes as wide as in-store operations and food waste [24,46], reverse logistics [47], or marketing and communication [48,49]. Recently, de Moraes et al. [46] conducted a systematic review to map the causes of food waste to practices adopted by retailers to reduce the magnitude of that phenomenon, whereas Huang et al. [50] developed a systematic literature review to understand how food retailers from 27 different countries deal with food waste both internally and externally. However, the topic still offers large room for exploration.

While there are some older works on the strategies and effect of the move towards green retailing [51,52], the recent nature of the interest in this research topic is patent in the numerous works focused on understanding and reporting on the causes and reduction practices of sustainability problems [16,46,53,54].

Some of the greatest operational challenges faced by retailers are related to the need to ensure adequate on-shelf availability, suitable inventory levels, and acceptable amounts of product shrinkage. These goals are actually interdependent among each other [27] and strongly depend on how effective the processes of a store are designed and executed. For these reasons, many retailers have been attempting to adopt operational excellence programs based on lean principles, methods, and tools to improve performance, productivity, and customer satisfaction [55–57]. While the origins of lean management are traced to the Toyota production system (TPS) [58], over time, it has seen several transformations [59] and applications [60], establishing it as a wide-ranging management system [61] that can be effectively adopted to eliminate wastes in any organization or industry [62–65]. Examples in the retail industry include how lean principles and concepts are successfully applied in 7-Elevens in Japan [66]. The promotion of a mindset at Tesco's supply chain and store operational flows [67,68], and the building of a lean continuous improvement (Kaizen) culture, was built at Amazon [69]. Regarding this topic, it is to highlight the research conducted by Domingo [70], who classifies OOS on shelves, in the context of lean, as waste in retailing.

In recent years, the concept of lean retail has been popularized—gaining naming variations that include "lean logistics", "lean distribution", and "lean consumption" [55,70]. Sowards [71] describe how lean can be applied to improve the productivity of a shop, attending to the continuous identification and elimination of the seven types of waste. Evans and Lindsay [72] report a Kaizen event conducted at the retail services of Magnivision to investigate problems that continuously plagued employees. Jaca et al. [73] described how a distribution center enhanced its productivity by removing inefficiencies at the warehouse. Noda [56] explained the sustained adoption of standardized work and process improvement practices based on lean principles by a mid-size Japanese retailer that sold foods, consumables, apparel, and general merchandise goods.

### 2.2. Retail Food Waste, Lean and Sustainability

Within the fresh products markets, perishability is a critical issue. As reported by Kor et al. [74], approximately 45% of all fruits and vegetables, 35% of fish and seafood, 30% of cereals, and 20% of meat and dairy products are wasted by suppliers, retailers, and consumers every year. In the retail management disciplines, the issue of retail food waste has been related to the "rate of shrinkage", a performance indicator that represents the gap between inventories and sales and is commonly used as an indicator of the performance of retail stores [1,75].

The management of perishable products is not only exposed to the stigma of food waste, but demands a high cost in its prevention [11]. Given their limited life span,

the freshness of perishable items is quickly lost with time. Every perishable item has a certain date before which it needs to be sold to the consumer. As a result, the handling and transportation of these products is subject to several requirements, strict norms, and sanitary regulations. On the sustainability side, their fast perishability—which affects storage conditions, transportation, order frequency and times, temperature control, packaging, and origin and traceability [76]—originates additional constraints and requirements. Grocery retailers are thus facing increasing pressure to tackle sustainability issues, promoting the transparency of their food supply chains and reducing food waste across them [77]. To Srivastava et al. [76], the fresh food retail industry is particularly vulnerable to supply chain risks.

In direct pursuit of a sustainability performance, or due to compliance, reputational gains, or as an opportunity to enhance business efficiency, retail companies are increasingly aware of the need to take actions to reduce product loss [11], including situations of food waste. Research in this scope has been conducted in various countries—where movements against food waste are emerging, asking the food chain actors, including retailers, for specific interventions to face this issue [78–81]. However, research on the effects of lean programs on sustainability metrics has been limited [82,83].

The prevailing literature describes lean as an approach that has a main objective of systematically identifying and eliminating waste in the organizational processes [84]. In fact, the entire lean toolbox is focused on the elimination of waste, purging all actions where there is no addition of value [85–87]. Food production takes a considerable part of the research on the relationship [88–90]. Another significant part of the studies on the balance between lean and green is focused on the supply chain, often influenced by approaches and perspectives previously used in the manufacturing sector [88,91,92]. Nevertheless, there is limited research on how to balance lean and green in in-store operations [93,94], and thus a gap is especially evident in the collection of empirical evidence. In this scope, we advance a case study showing the results that may be achieved by using lean methods and tools in pursuit of both the operational and sustainability performance.

Given that operational improvement efforts in the retail sector have frequently been addressed by the use of lean management, an opportunity presents itself to understand if and how lean may be used to tackle issues that pertain to both the operational and sustainability performance—and, as desired by this article, to explore the results that may be achieved in both dimensions when lean methods are applied to fresh products' markets.

Food waste is firstly and foremost a sustainability issue, since it generates significant economic, social, and environmental impacts [78], as is pointed out by the 2030 United Nations Agenda [46]. Nevertheless, the research conducted by Teller et al. [24] shows that the root-causes for the occurrence of food waste in the retail sector depend deeply on operational issues, such as the store format and product category, but also customer behaviors, demand variability, a poor efficiency within in-store operations, replenishment procedures, and high-demanding requirements for the quality of the products from both retail organizations and customers.

In the specific problem of food waste, the very basic principles of lean management provide a clear alignment with the challenges faced by fresh products' retailers. Retailers need to strike a balance between maximizing product availability on the shelves while minimizing the wastage of perishable products [35]. Factors such as a limited lifetime, high safety, and quality requirements, together with short lead time requirements, make them highly complex to manage [26]. In the fresh food categories, product deterioration is the biggest concern of food retailers, not only because it negatively affects the customer perception of quality, but also because of the monetary loss it creates [16].

The relationship between lean and green is thus balanced between benefits and mis-alignments. Sanchez-Rodrigues and Kumar [82] show, for example, that the implementation of a lean program in a particular company has been especially beneficial in food supply chains in terms of achieving a significant food waste reduction. However, the authors also highlight some misalignments, including superior vehicle usage, increased emissions, and

swollen costs due to fleet renewal towards electric vehicles, caused by JIT deliveries and tensions between time and stock and multiple deliveries within the same time window. Looking at the efforts used to reduce food waste, we found that metrics such as OOS are often used and may provide both operational and suitability gains. However, they may also present trade-offs—and isolating one single metric will offer just one piece in a broader picture [27]. In this sense, the adaptation of the "shrinkage" metric to include only situations of food waste due to poor handling, storage and display, exceeded expiration date, or similar factors offers a better opportunity to jointly tackle operational and sustainability issues.

## 3. Case Study

The case study focuses on one store of a multinational corporation in the retail sector located nearby Lisbon, in Portugal. With more than 70 stores in the country, the corporation employs around 9000 people. The hypermarket store is a medium-large format type with an area of around 7000 square meters. In 2019, the company was developing a strategic pilot initiative with the aim of implementing a lean green daily management program. This program aimed to improve operational key performance indicators, and, in parallel, to increase sustainability metrics. Due to their importance for the overall sustainability results, the improvement efforts targeted the markets under the fresh products department. The metrics studied in this particular study were the "shrinkage rate"—directly responsible for both food and operational waste—and, as a complementary metric, due to its strong correlation, the "out-of-stocks index".

### 3.1. Problem Statement

At the start of this study, the company faced poor performances in the shrinkage metrics, as well as in the product availability indicators. The term "shrinkage", in this context, is applied with a different perspective, going beyond its traditional scope of a measure of lost sales. In this work, it includes situations of food waste due to poor handling, storage and display, exceeded expiration date, and similar factors. Given our focus on sustainability, it does not include events of theft, nor accounting errors, which are otherwise normally considered in this metric.

The problems addressed by this study were not exclusively felt in the pilot store; they were also transversal to the majority of the corporation's stores. The shrinkage problem is particularly severe in most of the fresh product's markets.

### 3.2. Methodology

Given our interest in exploring the ability of lean tools and methods to improve both operational and sustainability metrics, a typically lean approach was followed. The methodology followed is outlined in Figure 1. It is coherent with the Gemba Kaizen event multi-stage approach described by Martin and Osterling [95] and Hamel [96]. The activities that took place during the Kaizen event execution followed the six-step structured-problem solving method provided by the A3 framework.

The event herein described was conducted in the cold meat market, which includes both the counter and as a self-service area. The whole project took approximately four and a half months to be completed. The specific steps involved in each of the three phases, depicted in Figure 1, as well as their duration, are detailed below:

1.  Kaizen Event Planning (duration: 2 weeks):

    a.  Definition of the event scope;
    b.  Selection of team members;
    c.  Gathering of relevant data;
    d.  Planning of the workshop;

2.  Kaizen Event Execution (duration: two-day workshop):

    a.  Ground rules, agenda and methodology;
    b.  Definition of the problem (step 1 of A3 Problem Solving);

  c. Characterization of the problem and determination of the baseline performance, or current condition (step 2 of A3 Problem Solving);

  d. Goal statement, or desired future state (step 3 of A3 Problem Solving);

  e. Root-causes analysis (step 4 of A3 Problem Solving);

  f. Definition and implementation of an action plan (step 5 of A3 Problem Solving);

  g. Definition of a control plan (step 6 of A3 Problem Solving);

3. Kaizen Event Follow-up—stabilization (duration: 4 months):

  a. Monitoring of the actions' effectiveness;

  b. Standardization, training and process stability.

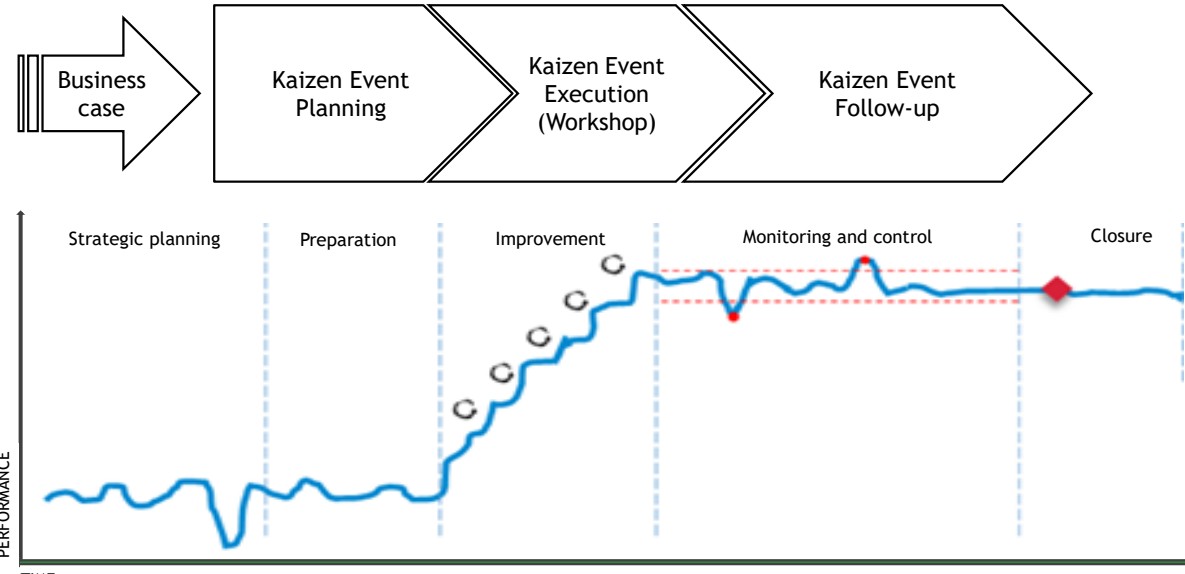

**Figure 1.** The stages of the Gemba Kaizen event and their relationship with the phases of process management and its performance.

### 3.3. Kaizen Event Planning

Data from the first quarter of 2019 revealed that the overall product shrinkage in the store that served as a pilot was higher compared with the same period of the previous year. In addition to this, a negative trend asserted itself over several months in the food waste indicators. It was thus considered important to conduct a set of Kaizen events in order to decrease the store shrinkage rate, a metric that relates the total monetary value lost due to food waste factors with the overall store sales.

The shrinkage rate usually correlates with the out-of-stocks (OOS) index. In facts, stockouts tend to increase when products that would be available for sale have to be removed from the shelves due to inadequate conditions. For this reason, the project was also a good opportunity to increase the product availability, so the team decided that it was important to consider the OOS index too.

As part of the preparation for the workshop stage, the facilitator of the event, together with staff from the store, gathered quantitative and qualitative (through direct observations) data, not only to be able to estimate baseline performances, but also to support a careful diagnosis of the current situation of the processes involved in the shrinkage creation, as well as to help the team with factual evidences during the root-cause analysis.

The project directly involved the director of the store, who acted as a sponsor; a member of the lean management team of the company, who provided the expertise in the principles, methods, and tools; the manager of the cold meat market, who participated as process owner in the project; and all of the four employees from the permanent operational staff belonging to that market. The store operators of the market and their manager were invited to participate in the workshop, whose duration was scheduled to last two days. An

experienced member from the internal lean team was selected to facilitate the workshop. The agenda, objectives, and outcomes for the event—both in terms of the sustainability and operational performance—were defined and communicated to participants and interested parties along with other relevant resources; in particular, the material needed to conduct the workshop and other logistics matters.

### 3.4. Kaizen Event Execution

The methodology followed in the workshop was completely aligned with the structured problem-solving mindset. The A3 report, exhibited in Figures 2 and 3, summarizes the steps followed by the team during the two-day event. The detail of each sequence of steps are presented in the following subsections.

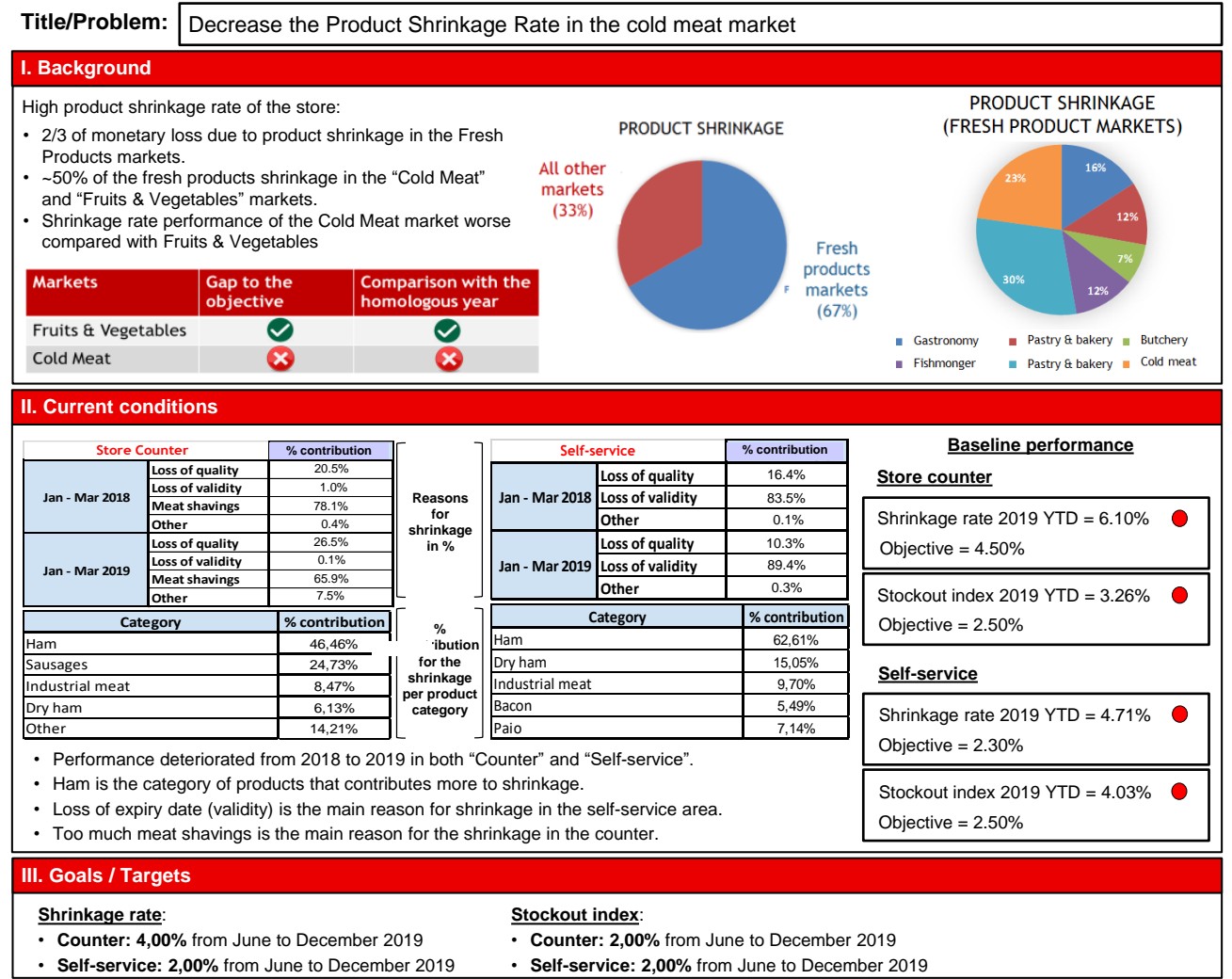

**Figure 2.** A3 problem-solving for day 1 of the Kaizen workshop.

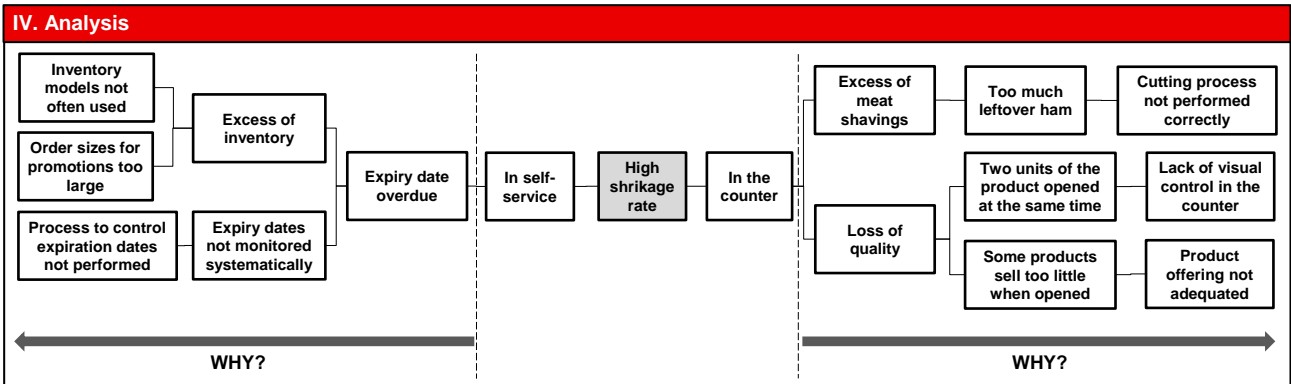

**Figure 3.** A3 problem-solving for day 2 of the Kaizen workshop.

### 3.4.1. Definition of the Problem

Collected data allowed for the store to understand that the highest levels of product shrinkage were in its fresh products markets, particularly in two areas: fruits and vegetables (F&V) and cold meat. The F&V market performed satisfactorily, since the shrinkage results were better than both the objective and in comparison with the previous year. On the contrary, the store performed poorly regarding this food waste indicator in the cold meat market, with a shrinkage rate far from the stated objective, as well as from the previous year's performance. These facts were recorded in advance in the "Background" field of the A3 report (Figure 2). The scope was therefore defined: decrease the shrinkage rate in the cold meat market.

### 3.4.2. Characterization of the Problem and Estimation of the Baseline Performance

Despite the shrinkage rate being the main indicator to be impacted by this event, it was also considered important by the team to follow the evolution of the stockouts index, which measures the percentage of non-available products on the shelves, because these indicators

are usually correlated with each other. The baseline for these KPIs was determined, and, for both metrics, the team realized a large gap between the results and objectives.

As one can observe in Figure 2, the cold meat market comprises two areas:

1. Counter;
2. Self-service.

Data from 2019 led to the conclusion that the end of the expiration date was identified as having by far the largest impact on the shrinkage value reported in the self-service area. The same conclusion could be drawn based on the data values of the same period of the previous year. Furthermore, "ham" was the product category that contributed the most for the overall shrinkage in both areas: self-service and counter.

In the counter, the excess of meat shavings was identified as having the most impact on both the food waste and monetary loss. Meat shavings correspond to resold trimmings that result from slicing the ham, including the remaining piece after cutting. They contribute to the reduction in the overall food waste; however, the fact that the product is sold as shavings and not as ham implies an economic loss for the store. A loss of quality is also a relevant reason for waste in the counter: because some products do not have sufficient demand, they are likely to overpass the required due time once opened.

### 3.4.3. Setting of Improvement Goals

The third step was the setting of SMART (specific, measurable, achievable, relevant to business, time-bounded) goals for both KPIs. The goals, included in the third room of the A3 exhibited in Figure 2, were aligned with the annual objectives established and communicated by the enterprise to employees.

### 3.4.4. Root-Causes Analysis

A root-cause analysis exercise was performed during the workshop to determine the underlying reasons for the shrinkage occurrence in both the counter and the self-service area. By consecutively asking "why", it was possible to construct a branched tree diagram of causes (Figure 3). Each possible deployed sub-cause was only validated and then only recorded if evidence (e.g., through "go and observe" walks) and factual data proved their relevance. The conjoint exercise in the workshop regarding the determination of potential root causes and their prioritization benefited from the work carried out during the Kaizen event planning stage, where quantitative and qualitative data were gathered in advance and a preliminary diagnosis could be performed.

In the counter, it was possible to gather evidence that could confirm the following root causes:

- The cutting process is not performed correctly, which tends to generate an excess of cold meat shavings. It was possible to verify that the way the ham cutting process was usually performed did not follow the best practices, thus producing an excess of trimmings;
- A lack of existing visual controls in the counter do not prevent the two pieces of the same product reference from being inadvertently opened;
- The range of products offered in the counter is not adequate. It was found that there is around a dozen products whose level of demand is quite low. The risk that these products, once opened, may have their quality degraded is significant. This means that such products should not have been available at the counter.

The root-causes determined in the self-service area were the following:

- Poor inventory management practices; in particular, two situations:
  - ○ Available inventory models are not used or not correctly utilized by the market manager;
  - ○ Very large order sizes when product promotions occur. It was found that, very often, the quantities of products ordered for promotion events were too high, resulting in too high inventory levels;

- The envisaged process to control the products' expiry dates were not consistently performed, mainly due to a lack of an effective daily planning and management.

### 3.4.5. Definition of Planned Countermeasures (Action Plan)

All of the root-causes determined in the previous step were validated by the team members participating in the workshop. It allowed for people to appropriate the improvement actions or countermeasures that were defined to respond to each root-cause. They are described in the fifth section of the A3 report. For example, a visual standard (Figure 4) was developed to assist the personnel at the counter in better performing the ham-cutting process. In addition to this, the standard also specified the acceptable weight of ham that should be left.

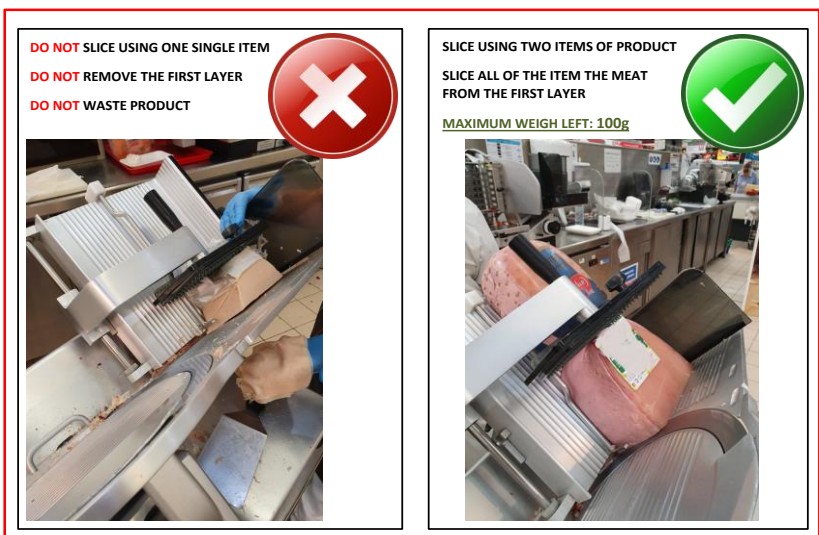

**Figure 4.** Visual standard for the "ham-cutting" operation created during the Kaizen event.

### 3.4.6. Definition of a Control Plan and Planning for Follow-Up Actions

Finally, a set of follow-up actions was defined, where key metrics would be measured, monitored, and evaluated on a daily or weekly basis. Due to the fact that the shrinkage rate is a lagging indicator, and is only available on the following day, the team tried to define a leading indicator to track the performance. For example, in the counter, the team monitored during the first weeks of the follow-up the percentage of cut ham items whose weight was above the acceptable weight indicated in the standard illustrated in Figure 4. This also allowed the team to confirm that the leftover ham was a relevant cause for the shrinkage rate, since the higher the percentage of items above the upper limit, the higher the shrinkage.

### 3.5. Kaizen Event Follow-Up

The follow-up actions were determined and put in place to verify if the countermeasures produced the expected result. This phase also aimed to maintain the new procedures and routines in order to stabilize processes. Run charts were used to visually monitor the evolution of KPIs on a daily basis. Figure 5 illustrates the chart used to monitor the accumulated overall shrinkage rate generated in the counter, where it was possible to visualize a positive trend towards the objective. Similar charts were created and used to control the evolution of the shrinkage rate regarding the main product categories: ham and sausages.

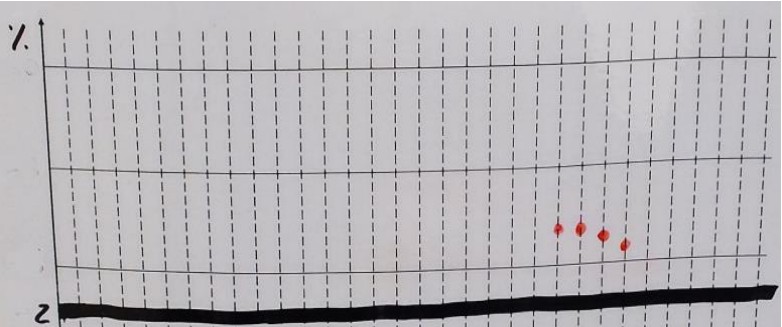

**Figure 5.** Run chart to monitor the overall shrinkage rate in the counter.

The already-in-place shrinkage control procedures contemplates the measurement and recording of the amount/weight of wasted product, stratifying by product category and by the type or cause of the shrinkage.

The general trend of the indicators was positive. It was interesting to notice that stockouts also decreased, thus meaning that shrinkage reduction was not achieved at the expense of a lack of product; in fact, quite the contrary. Figure 6 shows the evolution of the shrinkage rate and of the stockouts index in the counter. On average, the percentage values for the shrinkage rate were reduced to around a half, thus much more positively impacting the reduction in the food waste generated by this market. Moreover, the countermeasures were also revealed to be effective in reducing the out-of-stock percentage, which was decreased by one third. The results in the self-service area were similar.

**Objective= 2,00%**

| | Period | Shrinkage rate [%] | | Shrinkage rate in 2018 [%] | Stockouts index [%] |
|---|---|---|---|---|---|
| 2019 | Until May 2019 | 4.71% | 🔴 | 3.82% | 4.03% |
| | 1-14 June 2019 | 4.18% | 🟡 | 5.24% | 3.55% |
| | 15-30 June 2019 | 2.01% | 🟡 | 5.49% | 3.50% |
| | 1-14 July 2019 | 1.27% | 🟢 | 4.07% | 3.06% |
| | 15-31 July 2019 | 2.33% | 🟡 | 3.63% | 2.96% |
| | 1-14 August 2019 | 1.35% | 🟢 | 4.80% | 3.14% |
| | 15-31 August 2019 | 4.43% | 🔴 | 3.00% | 2.41% |
| | 1-14 September 2019 | 1.74% | 🟢 | 4.19% | 2.64% |
| | 15-30 September 2019 | 2.75% | 🟡 | 4.97% | 2.82% |
| | 1-14 October 2019 | 1.54% | 🟢 | 4.19% | 2.60% |

🟢    Comply with the objective and it is better when compared with the previous year

🟡    Do not comply with the objective but it is better when compared with the previous year

🔴    Do not comply with the objective and it is worse when compared with the previous year

**Figure 6.** Evolution of the two main KPIs in the counter.

From Figure 6, one can also notice a deterioration of the shrinkage rate performance during the summer period in August. It reveals a higher variability during this period, mainly due to the vacation period of some of the permanent staff and their replacement by temporary and untrained employees who are hired during this period. In order to sustain the achieved performance results, a set of procedures were standardized and the people from the cold meat market were trained in these standards, including the temporary personnel.

The results demonstrated the overall effectiveness of the countermeasures in reducing the food waste measured by the shrinkage rate and, at the same time, decreasing the proportion of stockouts, hence contributing to increased sales. This integrated perspective provides an interesting notion of the strong relationship existing in food retail between social and environmental sustainability and economic and business sustainability.

## 4. Discussion and Conclusions

This paper described the case study of a Gemba Kaizen event conducted at a food retail store in Portugal with the main purpose of reducing the "shrinkage rate" in the cold meat market, a sustainability indicator related to food waste. As was explained in the Kaizen event planning stage, a secondary aim concerned the diminishing of OOS measured by an operational indicator called the "stockouts index". The event comprised three stages—preparation, improvement workshop, and follow-up—and it was conducted as part of a daily Kaizen program that was being implemented in the store, which served as a pilot for a broader and strategic project of the company called a "lean store".

Improvement actions were defined and implemented in the two areas that comprise the cold meat market (self-service and counter) after determining the root-causes for the shrinkage problem. During the weeks that followed the Kaizen workshop, the pilot store was able to reach the top-ranking position regarding the shrinkage indicator in the cold meat market, among all of the stores that the retail company own in Portugal. The national coordinator for the cold meat market has estimated overall savings of more than EUR 100,000 a year if the measures introduced by this project are adopted country-wide.

The main performance indicator classified in the sustainability category monitored by the company is the percentage of cardboard and plastic waste resulting from the replenishment operations that are recycled. Typically, the shrinkage rate was not regarded as an indicator of sustainability, but mainly as a metric that is indicative of the amount of unsold product. This project introduced the theme of lean green within the company. One of the things that attracted attention from this initiative was that the reduction in the shrinkage rate not only contributed to decreasing the amount of wasted food, but also contributed to diminishing the OOS occurrences, thus contributing to increase the sales of the store. For these reasons, and from a business point of view, it was possible for managers to establish a linkage between sustainability and the financial results of the company.

Two foundational success factors were revealed to be critical for the outcome of this project: firstly, the visible commitment from the leadership of the store, and, secondly, the involvement of the operational staff from the very beginning during the pre-diagnosis activities, which allowed them to understand and be aware of the importance of reducing the amount of wasted food, while contributing to increase sales.

## 5. Limitations and Future Research

The case study described in this paper was conducted at a single store with a specific format dimension. The operations and other considerations made regarding the cold meat market described for the pilot store are very applicable to other hypermarket formats too; however, we do recognize the situation as a limitation of the study. In addition, the researchers were not able to conduct a study on the reduction in the shrinkage rate in other relevant markets from the fresh product's department. To benchmark the learning derived from the lean case herein presented, there is a need to extend this research to another store format in order to validate the applicability of the lean and green methodologies, not only in the cold meat areas, but also in other fresh products' markets.

**Author Contributions:** Conceptualization, P.A.M. and A.M.C.; methodology, P.A.M. and A.M.C.; formal analysis, P.A.M. and A.M.C.; investigation, P.A.M.; resources, P.A.M. and A.M.C.; writing—original draft preparation, P.A.M. and A.M.C.; writing and editing, P.A.M. and A.M.C.; project administration, P.A.M. and A.M.C.; funding acquisition, J.O.S. All authors have read and agreed to the published version of the manuscript.

**Funding:** This research was funded by the HORIZON 2020 EU Programme PhytoAPP project, grant number 101007642 H2020-MSCA-RISE-2020.

**Institutional Review Board Statement:** Not Applicable.

**Informed Consent Statement:** Not Applicable.

**Data Availability Statement:** Restrictions apply to the availability of the data.

**Conflicts of Interest:** The authors declare no conflict of interest.

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
