# Peer review of "Improving Operational and Sustainability Performance in a Retail Fresh Food Market Using Lean: A Portuguese Case Study"

_sustainability, doi:10.3390/su14010403_

Round 1

Reviewer 1 Report

This is a good paper, well written and well-structured and with a clearly expressed literature review.

This are some recommendations for the authors to further improve the paper:

  1. The literature review although relevant is too long, it occupies half of the length of the paper and gives less space to better discuss the case study.

  1. Information that could help to deepen the case study analysis and to extend it to another market store is namely:
  • The type and size of the supermarket
  • how many employees/ operational staff where involved? all the employees of those segments?
  • How much time it took to develop the 6 phases of the project?

  1. Figure 1 as it is does not allow these conclusions to be drawn “As depicted in figure 1, each paper was classified according to each combination of both dimensions. The authors concluded that more than two thirds of all papers fall in the “inventory management decision” category; and within this category, that more than 95% focus on the “Uncertainty”, “Perishability”, and “Availability” themes.” (lines 133-136)

It is not clear what the figure adds in terms of information as all cells are indicated with an arrow but no values are provided and it is not possible to distinguish the relevance of each dimension or how many papers have been classified in relation to each one.  For the figure to have added value, the cells should be filled with more precise information, for example, the titles of the papers that were identified in each one of them.

If this is not possible, the figure as it is should be removed.

  1. In the discussion it is clear how the project implementation contributed to reduce the shrinkage rate but no clear explanation was given to how “it contributed to diminish the OOS occurrences, thus contributing to increase the sales of the store.” As well as how they related to each other.

“One of the things that attracted attention from this initiative was that not only the reduction of the shrinkage rate contributed to decrease the amount of wasted food, but also that  it contributed to diminish the OOS occurrences, thus contributing to increase the sales of  the store.” (lines 540-543)

  1. Is there any explanation for the variability of the shrinkage rate (Figure 7)?

Minor changes

Line 89 “being followed by the discussion of the results and a summary of the conclusions”

Line 344 “with limited research on the/how?  to balance Lean and Green in in-store operations”

Line 537 “Typically, the shrinkage rate was not regarded as  an indicator of sustainability but mainly as a metric as a metric indicative of the amount”

Author Response

Dear Reviewer.

Please find attached to this message the  file with the details of the review of the paper.

Thank you

Reviewer 2 Report

This work sheds light on the application of improvement steps aiming at Lean-Green procedures in the cold meat section of a store. The authors did not follow the guidelines for this journal, therefore, is of utmost importance that they do so, to continue the reviewing process.

This is an interesting topic and I get the idea of using a measure like a shrinkage rate, although I have some doubts on the way shrinkage rate can be used as a Lean Green indicator, since is purely related to the quantity of material that is not sold by the company. In my opinion, the "green" part of the study was somehow neglected. Can the authors elaborate on that?

This work showed that the proposed model helped in reducing the consumption of resources aiming at improving their environmental performance.  However, as pointed out by the authors,  this study must be adopted in other stores to validate this model of operation.

Since it is very natural to encounter resistance from employees implementing new business initiatives, the strategy behind the establishment of root causes can be revised in order to achieve a quicker overturn in the results after implementation. Can the authors give some insight on how root causes were defined?

Introduction
L48. Please elaborate on the term "shrinkage" and how it applies here.
L59. What do you mean by the "research trend is still emerging"?

Literature Review 
Since this is not a Review article, this section must be revised and incorporated into the introductory text. Also, the article sections should be revised to fit the manuscript sections according to the journal guidelines.

L152. Is limited lifetime different from shelf-life?

Theoretically, everything seems to be in place, the problem definition and the way the Kaizen planning was done as well. However, the bridge between the hypothesis and results is not there. The results must be better described and discussed overall. It seems that this section is fairly incomplete and only touches on some aspects. 

The countermeasures and follow-up are essential and should be limited to minimize the monitoring work. Were there any constraints observed? What was done to minimize them?

L468. Are there any different shapes of the product? Was there any conclusion gathered on the effect of the shape of the sliced material on the shrinkage rate?

At which level were sustainability categories measured during the implementation?

Author Response

Dear reviewer.

Please find attached to this message a  file with the details of the changes introduced in the paper.

Thank you

Round 2

Reviewer 2 Report

The authors provided answers for most of the asked questions.

L468. In regard to the shape of the items, was there any measure in place to measure shrinkage disparities, or the tested samples were equal?

The text on Section 3.4 should explain better the workflow presented in Figure 2.

Author Response

Dear reviewer.

We are attaching the answers to each comment of the review. We believe clarifitions and improvements were introduced to meet the comments.

Thank you,

The authors
